# Unveiling a New Perspective on Distinguishing Omicron Breakthrough Cases and Postimmune COVID-19-Naive Individuals: Insights from Antibody Profiles

Shihan Zhang,[a,b] Chen Dong,[c] Qian Zhen,[d] Chao Shi,[e] Hua Tian,[c] Chuchu Li,[c] Xiaoxiao Kong,[c] Qigang Dai,[c] Haodi Huang,[c] Aidibai Simayi,[a,b] Fengcai Zhu,[c,f,g] Yawen Xu,[h] Jianli Hu,[c] Ke Xu,[c] Liling Chen,[i] Changjun Bao,[c,j] Hui Jin,[a,b] Liguo Zhu[c,f,g,k]

[a]Department of Epidemiology and Health Statistics, School of Public Health, Southeast University, Nanjing, China

[b]Key Laboratory of Environmental Medicine Engineering, Ministry of Education, School of Public Health, Southeast University, Nanjing, China

[c]Department of Acute Infectious Disease Control and Prevention, Jiangsu Provincial Center for Disease Control and Prevention, Nanjing, China

[d]Department of Acute Infectious Disease Control and Prevention, Changzhou Center for Disease Control and Prevention, Changzhou, China

[e]Department of Acute Infectious Disease Control and Prevention, Wuxi Center for Disease Control and Prevention, Wuxi, China

[f]National Health Commission (NHC) Key Laboratory of Enteric Pathogenic Microbiology, Jiangsu Provincial Center for Disease Control and Prevention, Nanjing, China

[g]Key Laboratory of Infectious Diseases, School of Public Health, Nanjing Medical University, Nanjing, China

[h]Yangzhou Center for Disease Control and Prevention, Yangzhou, China

[i]Suzhou Center for Disease Control and Prevention, Suzhou, China

[j]Jiangsu Province Engineering Research Center of Health Emergency, Nanjing, China

[k]Jiangsu Key Lab of Cancer Biomarkers, Prevention and Treatment, Jiangsu Collaborative Innovation Center for Cancer Medicine, Nanjing Medical University, Nanjing, China

Shihan Zhang, Chen Dong, Qian Zhen, and Chao Shi share first authorship. Author order was determined by communication.

**ABSTRACT** In the situation of mass vaccination against COVID-19, few studies have reported on the early kinetics of specific antibodies (IgG/IgM/IgA) of vaccine breakthrough cases. There is still a lack of epidemiological evidence about the value of serological indicators in the auxiliary diagnosis of COVID-19 infection, especially when the nucleic acid results were undetectable. Omicron breakthrough cases post-inactivated vaccination ($n = 456$) and COVID-19-naive individuals with two doses of inactivated vaccination ($n = 693$) were enrolled. Blood samples were collected and tested for SARS-CoV-2 antibody levels based on the magnetic chemiluminescence enzyme immunoassay. Among Omicron breakthrough cases, the serum IgG antibody level was 36.34 Sample/CutOff (S/CO) (95% confidence interval [CI], 31.89 to 40.79) in the acute phase and 88.45 S/CO (95% CI, 82.79 to 94.12) in the recovery phase. Serum IgA can be detected in the first week post-symptom onset (PSO) and showed an almost linear increase within 5 weeks PSO. Compared with those of breakthrough cases, IgG and IgA titers of the postimmune group were much lower (4.70 S/CO and 0.46 S/CO, respectively). Multivariate regression showed that serum IgG and IgA levels in Omicron breakthrough cases were mainly affected by the weeks PSO ($P < 0.001$). Receiver operating characteristic ROC0 curve analysis showed that the area under the curve (AUC) was 0.744 and 0.806 when the cutoff values of IgA and IgG were 1 S/CO and 15 S/CO, respectively. Omicron breakthrough infection can lead to a further increase in IgG and IgA levels relative to those of the immunized population. When nucleic acid real-time PCR was negative, we would use the kinetics of IgG and IgA levels to distinguish the breakthrough cases from the immunized population.

**IMPORTANCE** This study fills a gap in the epidemiological evidence by investigating the value of serological indicators, particularly IgG and IgA levels, in the auxiliary diagnosis of COVID-19 infections when nucleic acid results are undetectable. The findings reveal that among Omicron breakthrough cases, both IgG and IgA antibody levels exhibit significant changes. Serum IgG levels increase during the acute phase and rise further in the recovery phase. Serum IgA can be detected as early as the first week post-symptom

Address correspondence to Hui Jin, jinhui_hld@163.com, or Liguo Zhu, zhulg@jscdc.cn.

The authors declare no conflict of interest.

onset (PSO), showing a consistent linear increase within 5 weeks PSO. Furthermore, receiver operating characteristic (ROC) curve analysis demonstrates the potential of IgG and IgA cutoff values as diagnostic markers. The study's conclusion underscores the importance of monitoring IgG and IgA kinetics in distinguishing Omicron breakthrough cases from vaccinated individuals. These findings contribute to the development of more accurate diagnostic approaches and help inform public health strategies during the ongoing COVID-19 pandemic.

**KEYWORDS** SARS-CoV-2, specific antibodies, Omicron breakthrough infection, vaccinated, early kinetics

As of February 2023, SARS-CoV-2 had caused more than 700 million confirmed COVID-19 cases and over 6 million deaths globally, and more than 13.2 billion vaccine doses had been administered (1). The Omicron variant has been detected in several countries and regions around the world (2). SARS-CoV-2 vaccines are effective in reducing the risk of symptomatic SARS-CoV-2 infection and progression to severe COVID-19 (3). However, as antibody titers wane over time, the vaccine effectiveness decreases (4, 5). This provides suboptimal protection against emerging variants with immune escape potential, such as Omicron, leading to breakthrough infections and reinfections (6–10).

As vaccination for SARS-CoV-2 becomes increasingly available and the number of breakthrough cases continues to increase, it will be useful to understand the kinetics of antibodies in postvaccination and breakthrough infection populations. It is also worth exploring whether there are differences between the characteristics of humoral immunity in these two representative populations and in those of previously studied naturally infected individuals. For example, Cheng et al. monitored BBIBP vaccine-induced virus-specific antibody levels (SARS-CoV-2-IgA/IgM/IgG) over multiple time points and showed that serum antibody levels decreased over time (11). Further, Ma et al. reached a similar conclusion that the levels of receptor-binding domain (RBD) antibodies and IgG peaked 1 month after the second injection, while IgM and IgA levels remained consistently low throughout the course of vaccination (12). For naturally infected individuals (those without prevaccination), previous studies have shown that antibodies caused by SARS-CoV-2 appear 3 days after the onset of symptoms or 1 week after infection (13, 14). Our team also suggested in a previous study that, at a median follow-up of up to 15.6 months in a naturally infected population, 80.19% of patients were IgG positive and 20.44% were IgM positive (15). Through an extensive literature search, Koutsakos et al. indicated that immune recall following breakthrough infection varies in timing and magnitude. In addition, Omicron breakthrough infection elicits less extensive immune recall than Delta (16). Yadav et al. assessed the sera of naive, recovered, and breakthrough patients vaccinated with Covaxin for their neutralizing ability against the Omicron variant (17).

Important aspects of humoral immunity, such as the kinetics of IgG, IgM, and IgA over time after vaccination or post-symptom onset/infection (PSO), were neglected. There is still a lack of epidemiological evidence regarding the value of serological indicators in the auxiliary diagnosis of COVID-19, especially when nucleic acid results are undetectable. Therefore, a case-control study was conducted using Omicron breakthrough cases with inactivated vaccine and COVID-19-naive individuals with two doses of inactivated vaccine as the control group in Jiangsu Province, China. Our aim was first to capture the precise kinetics of specific antibodies in Omicron breakthrough cases in the acute and recovery phases and to compare them with those of the postimmune population. Second, we explored whether the quantitative levels of IgG/IgM/IgA played a role in distinguishing between these antibodies induced by vaccine-only or vaccine-infection dual function. Third, we attempted to determine the exposure time of infected individuals during a large epidemic of the Omicron variant.

## RESULTS

**Demographic and clinical characteristics of study participants.** The samples included in this study comprised 308 sera from the acute phase of Omicron breakthrough cases, 148 sera from the recovery phase, and 693 sera from the uninfected population that received 2

**TABLE 1** Demographic and clinical characteristics of study subjects

| Characteristic | Omicron breakthrough cases (samples) | | Postimmune population (samples) | P value[a] |
| | Acute phase | Recovery phase | | |
| --- | --- | --- | --- | --- |
| Total (n) | 308 | 148 | 693 | |
| Gender (n) | | | | <0.001 |
| Male | 188 (61.04) | 87 (58.78) | 200 (28.86) | |
| Female | 120 (38.96) | 61 (41.22) | 493 (71.14) | |
| Age, median, yrs (IQR) | 46 (32.5–54) | 46 (28–54) | 47(36–63) | <0.001 |
| Age (yrs) | | | | <0.001 |
| <18 | 32 (10.39) | 20 (13.51) | | |
| 18–65 | 254 (82.47) | 117 (79.05) | 549 (79.22) | |
| >65 | 22 (7.14) | 11 (7.43) | 144 (20.78) | |
| Inactivated vaccination | | | | <0.001 |
| Fully vaccinated | 175 (56.82) | 75 (50.68) | 693 (100.00) | |
| Homologous booster immunization | 133 (43.18) | 73 (49.32) | | |
| Median interval between the last dose and symptom onset/blood collection (mo) (IQR) | 5.45 (2.18–7.33) | 4.22 (2.18–7.18) | 2.67 (1.82–4.67) | <0.001 |

[a]Chi-square test or Fisher's exact test as appropriate; $P < 0.05$ represents a significant difference.

doses of the inactivated vaccine. The demographic and clinical characteristics of the study participants are presented in Table 1. For breakthrough cases, the male-to-female ratio in the acute phase was 61.04% versus 38.96%, and that in the recovery phase was 58.78% versus 41.22%. For the postimmune population, the male-to-female ratio was 28.86% versus 71.14%. The average age of the three groups was around 45 years, but the age distribution was slightly different, with 80% of breakthrough cases concentrated in the 18- to 65-year age group, followed by 10% in the <18 age group, while there were no individuals younger than 18 years in the postimmune group, who were mainly (79.22%) in the 18 to 65 age group with 20.78% >65 years. More than 50% of the breakthrough cases providing sera completed full vaccination, less than 50% of the patients received homologous booster immunization, and 100% of the postimmune population that provided sera were fully vaccinated (comprising two doses of inactivated vaccine). The median interval between the last dose and symptom onset in the acute phase sample of Omicron breakthrough cases was 5.45 months (interquartile range [IQR], 2.18 to 7.33), and for those in the recovery phase, it was 4.22 months (IQR, 2.18 to 7.18). For the postimmune population, the median interval between the last dose and blood collection was 2.67 months (IQR, 1.82 to 4.67). There were statistically significant differences in the demographic and clinical characteristics between the two groups.

**Comparison of IgG/IgM/IgA positive rates and antibody levels in two groups.** As shown in Table 2, the IgG/IgM/IgA-positive rates showed statistically significant differences among the groups ($P < 0.001$). The serum IgG positivity rate was 83.77% in the acute phase of Omicron breakthrough cases, 97.97% in the recovery phase, and 76.19% in the

**TABLE 2** Comparison of IgG/IgM/IgA positive rates in three groups of subjects[a]

| Group | IgG | | | | IgM | | | | IgA | | | |
| | Positive (n) | Negative (n) | Positive rates (%) | P | Positive (n) | Negative (n) | Positive rates (%) | P | Positive (n) | Negative (n) | Positive rates (%) | P |
| --- | --- | --- | --- | --- | --- | --- | --- | --- | --- | --- | --- | --- |
| | | | | | | | | | | | | <0.001 |
| Patients-acute | 258 | 50 | 83.77 | <0.001 | 41 | 267 | 13.31 | <0.001 | 122 | 186 | 39.61 | <0.001[b,c,d] |
| Patients-recovery | 145 | 3 | 97.97 | <0.001[b,c] | 50 | 98 | 33.78 | <0.001[b,c] | 131 | 17 | 88.51 | |
| Postimmune population | 528 | 165 | 76.19 | 0.007[d] | 57 | 636 | 8.23 | 0.012[d] | 46 | 647 | 6.64 | |

[a]$P < 0.05$ represents a significant difference.
[b]Patients-acute, patients-recovery.
[c]Patients-recovery, postimmune population.
[d]Patients-acute, postimmune population.

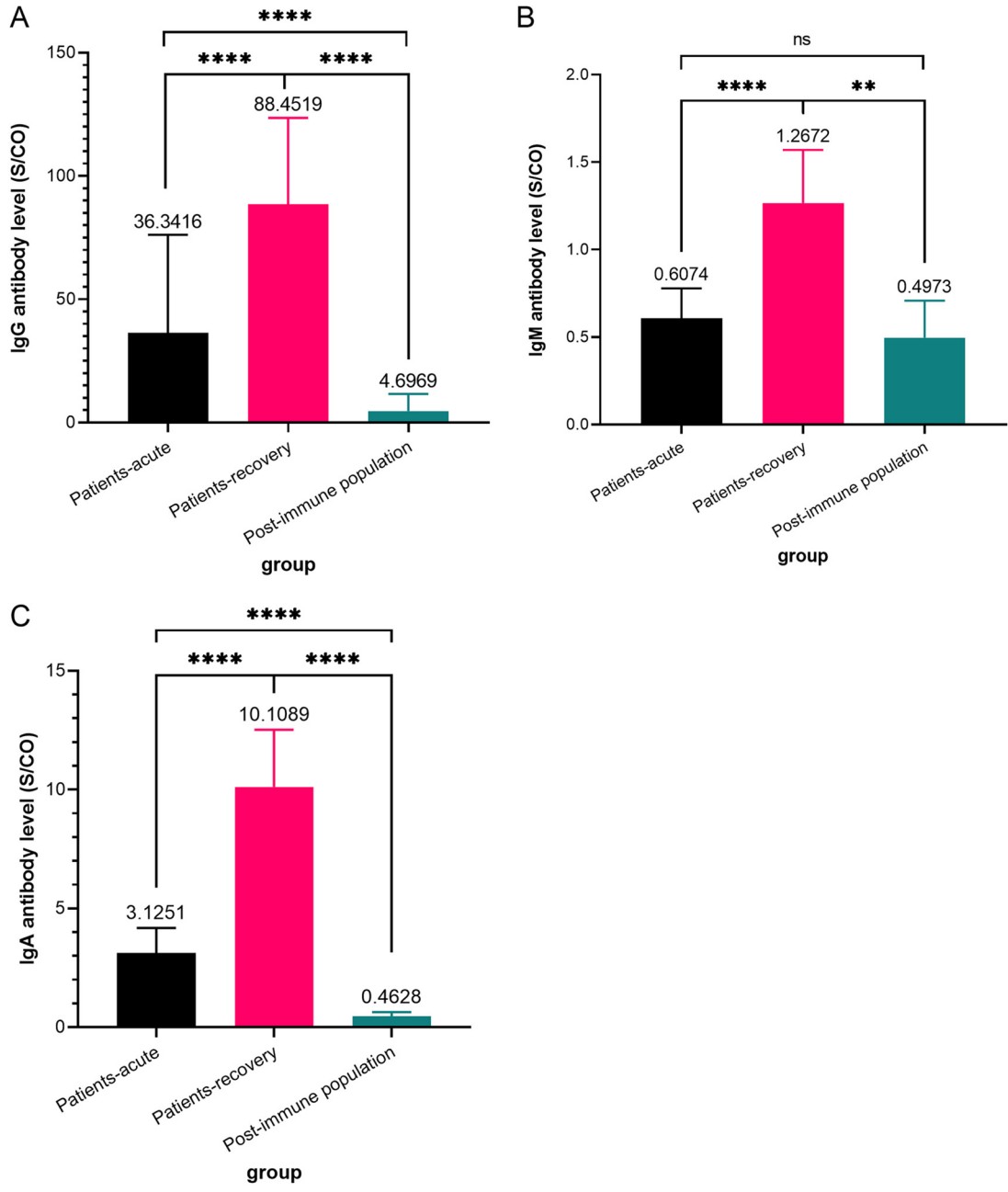

**FIG 1** Comparison of IgG/IgM/IgA antibody levels in three groups of subjects. (A) IgG antibody levels in the acute phase and recovery phase of Omicron breakthrough cases and the postimmune population. (B) IgM antibody levels in the acute phase and recovery phase of Omicron breakthrough cases and the postimmune population. (C) IgA antibody levels in the acute phase and recovery phase of Omicron breakthrough cases and the postimmune population. $P$ values were determined by applying a two-tailed Mann-Whitney U test. A $P$ value of $<0.05$ was considered statistically significant; ns, not significant; *, $P < 0.05$; **, $P < 0.01$; ***, $P < 0.001$; ****, $P < 0.0001$.

postimmune population. The serum IgM positivity rate was 13.31% in the acute phase of the Omicron breakthrough cases, 33.78% in the recovery phase, and 8.23% in the postimmune phase. The serum IgA positivity rate was 39.61% in the acute phase of the Omicron breakthrough cases, 88.51% in the recovery phase, and 6.64% in the postimmune phase. A comparison of IgG, IgM, and IgA antibody levels is shown in Fig. 1. The IgG antibody level in the acute phase sample of the Omicron breakthrough cases was 36.34 Sample/CutOff (S/CO) (95% confidence interval [CI], 31.89 to 40.79), that in the recovery phase was 88.45 S/CO (95% CI, 82.79 to 94.12), and that in the postimmune population was 4.70 S/CO (95% CI, 4.18 to 5.21) ($P < 0.001$). For IgM antibody levels, this measure was 0.61 S/CO (95% CI, 0.44 to 0.78) in the

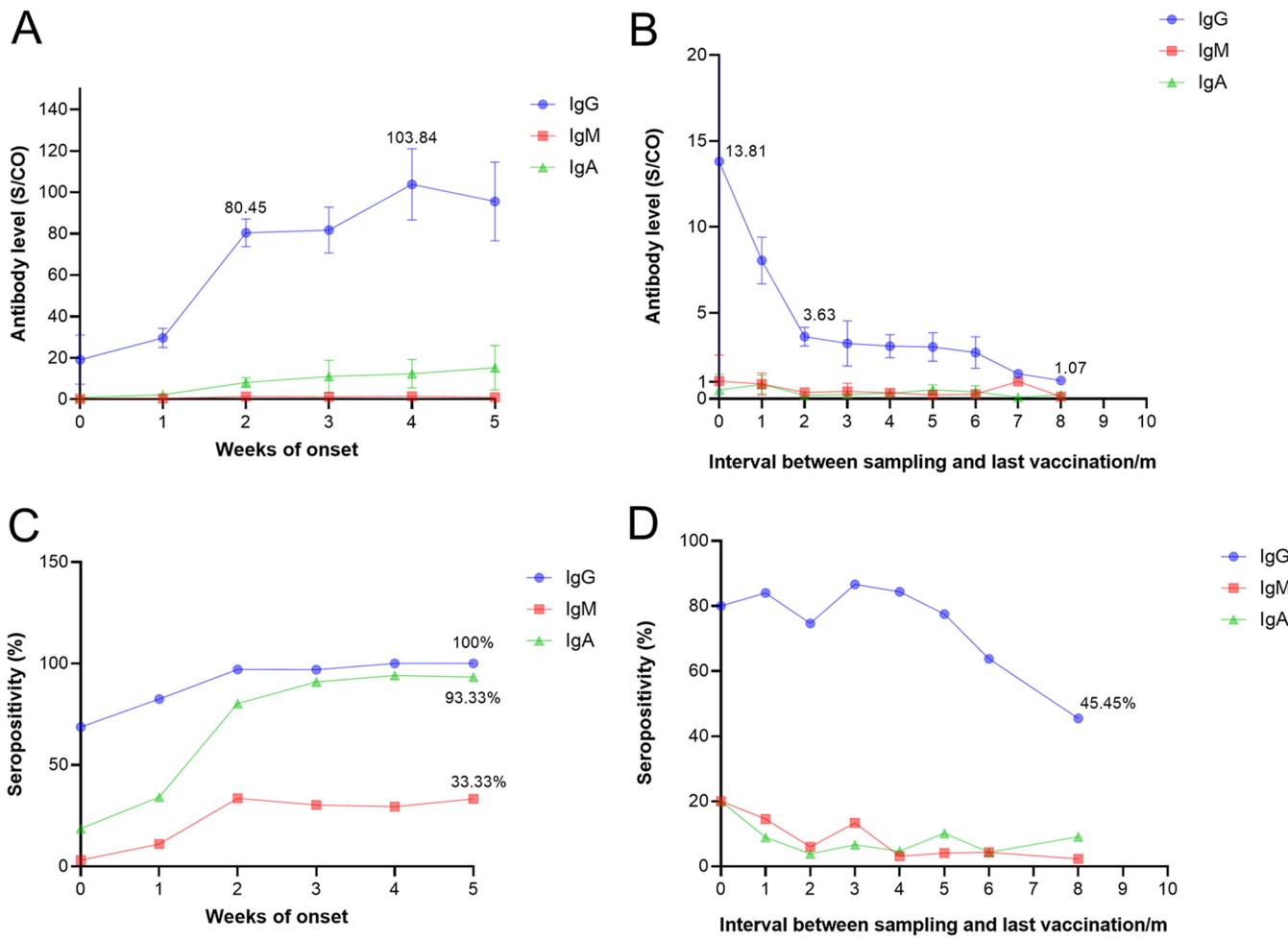

**FIG 2** Trends in specific antibody levels/positive rates in Omicron breakthrough cases and the postimmune population. (A) Time-varying kinetics of IgG/IgM/IgA antibody levels within 5 weeks after symptom onset in Omicron breakthrough cases. (B) Changes in antibody levels as the time interval between the last dose of vaccination and sampling increased in the postimmune population. (C) Kinetics of IgG/IgM/IgA seropositivity within 5 weeks after symptom onset in Omicron breakthrough cases. (D) Changes in IgG/IgM/IgA seropositivity as the time interval between the last dose of vaccination and sampling increased in the postimmune population.

acute phase of the cases, 1.27 S/CO (95% CI, 0.97 to 1.57) in the recovery phase, and 0.50 S/CO (95% CI, 0.29 to 0.71) in the postimmune population. We found no difference in IgM antibody levels between the acute phase of breakthrough cases and the postimmune population ($P > 0.05$). IgA antibody levels were 3.13 S/CO (95% CI, 2.08 to 4.17) in the acute phase of the cases, 10.11 S/CO (95% CI, 7.71 to 12.50) in the recovery phase, and 0.46 S/CO (95% CI, 0.29 to 0.64) in the postimmune population ($P < 0.001$). IgG and IgA titer levels were higher in the recovery phase of immune breakthrough cases than in the acute phase and even higher than in the postimmune healthy population, with statistical differences between the three.

**Time-varying kinetics of specific antibodies in two groups.** We found that the IgG level of breakthrough cases sharply rose to 80.45 S/CO in the second week PSO, followed by a fluctuating upward trend, and peaked at 103.84 S/CO in the fourth week PSO. In addition, we found that IgA was detected from the first week PSO and showed an almost linear increase within 5 weeks PSO. However, IgM levels were low and undetectable (Fig. 2A). The changes in antibody levels over time in the postimmune population showed a different pattern (Fig. 2B). Fig. 2B shows that the overall trend of IgG levels decreased, especially in the first 2 months after the last dose of vaccination. There was a sharp drop from 13.81 to 3.63 S/CO, after which it remained at a low level and finally reached 1.07 S/CO in the 8th month after full vaccination. The IgM and IgA antibody levels in the postimmune population fluctuated at low levels (<1 S/CO) throughout the study period.

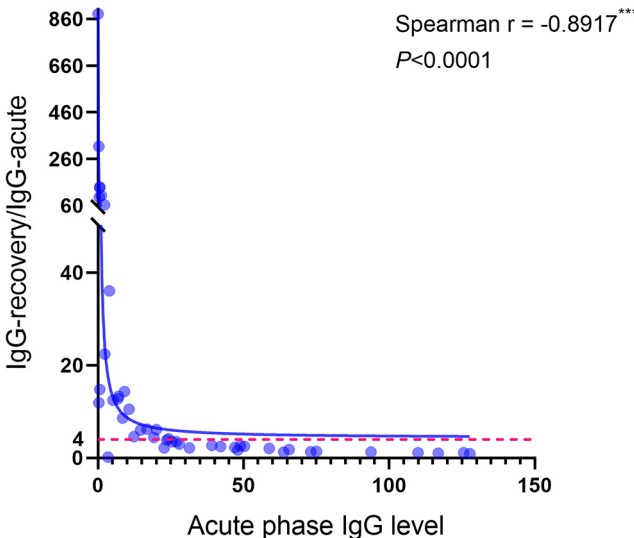

Spearman r = -0.8917[****]

*P*<0.0001

**FIG 3** Scatterplot between acute phase IgG levels and ratios of IgG recovery/IgG acute in Omicron breakthrough cases. The blue dots are independent samples selected to meet the conditions, the blue line is the nonlinear fitted curve, and the ratio of 4 on the vertical coordinate is drawn with a red dashed line. The correlation between the two indicators was calculated by Spearman correlation coefficient. A *P* value of <0.05 was considered statistically significant; ns, not significant; *, $P < 0.05$; **, $P < 0.01$; ***, $P < 0.001$; ****, $P < 0.0001$.

Omicron breakthrough cases showed a similar trend of IgG and IgA positivity, with an upward trend in the acute phase (within 2 weeks), followed by a relatively flat trend in the recovery phase (3 to 5 weeks), reaching 100% and 93.33%, respectively, by week 5 PSO (Fig. 2C). The IgG-positive rates in the postimmune population showed a fluctuating downward trend, from approximately 80% at the beginning to 45.45% at 8 months after vaccination. In addition, IgM and IgA positivity rates fluctuated by 20% (Fig. 2D).

**Comparison of IgG antibody levels between the acute and recovery phases of Omicron breakthrough cases.** For confirmed cases, one of the serological pieces of evidence required for a suspected case is a 4-fold or higher increase in IgG antibody titers in the recovery phase compared to the acute phase when they were not immunized. A total of 46 patients with two blood collections in the acute and recovery phases (14 days or more apart) were selected to explore the extent to which the above-described conclusion was met in confirmed cases. Details on the time of blood collection and IgG antibody levels for these individuals are provided in Table S1 in the supplemental material. Surprisingly, the ratio of IgG recovery/IgG acute gradually decreased as the level of IgG antibodies in the acute phase increased; therefore, we drew scatterplots of these two indicators and calculated Spearman's *r* ($-0.8917$, $P < 0.0001$; Fig. 3). We divided the acute IgG levels into four groups: <1 S/CO, 1 to 20 S/CO, 20 to 40 S/CO, and >40 S/CO. We calculated the proportion of IgG recovery/IgG acute ratios of four or more in each group, as well as the cumulative percentage. The proportion of ratios greater than 4 decreased as the level of IgG in the acute phase increased, with a total of 23 people (50%) meeting these criteria ($P < 0.05$; Table 3).

**TABLE 3** Comparison of acute and recovery IgG antibody titers in Omicron breakthrough cases[a]

| Acute phase IgG level (S/CO) | Recovery IgG level (S/CO) (median, IQR) | Kruskal-Wallis test | | IgG-recovery/IgG-acute (median, IQR) | Kruskal-Wallis test | | Ratios greater than 4 (%) |
|---|---|---|---|---|---|---|---|
| | | Chi-square | P | | Chi-square | P | |
| Overall | 94.06 (78.05, 119.13) | | | 4.00 (1.98, 13.53) | | | 23/46 (50) |
| <1 | 52.93 (9.32, 81.66) | 15.869 | 0.001 | 136.93 (14.79, 314.08) | 33.570 | <0.001 | 7/7 (100) |
| 1–20 | 87.63 (63.87, 119.12) | | | 12.43 (6.01, 22.41) | | | 14/15 (93.33) |
| 20–40 | 91.13 (75.81, 102.78) | | | 3.48 (2.43, 4.00) | | | 2/9 (22.22) |
| >40 | 118.76 (103.11, 120.17) | | | 1.42 (1.11, 2.23) | | | 0/15 (0) |

[a]$P < 0.05$ represents a significant difference.

**TABLE 4** Univariate and multivariate regression analysis of IgA antibody levels in 451 Omicron breakthrough cases[a,b]

| Variable | No. observed | Univariate regression | | Multivariate regression[a,b] | |
|---|---|---|---|---|---|
| | | B[c] | P | B | P |
| Total | 451 | | | | |
| Gender | | | | | |
| Male | 271 | Reference[d] | | Reference | |
| Female | 180 | −1.833 | 0.110 | −2.087 | 0.057 |
| Age (yrs) | | | | | |
| <18 | 52 | −1.395 | 0.433 | −1.825 | 0.285 |
| 18−65 | 341 | Reference | | Reference | |
| >65 | 58 | −0.709 | 0.676 | −1.908 | 0.240 |
| Booster dose | | | | | |
| Yes | 204 | 0.556 | 0.622 | | |
| No | 247 | Reference | | | |
| Wks since onset | | | | | |
| 0 | 32 | −1.275 | 0.551 | −1.364 | 0.523 |
| 1 | 217 | Reference | | Reference | |
| 2 | 137 | 5.931 | **<0.001** | 6.016 | **<0.001** |
| 3 | 33 | 8.819 | **<0.001** | 9.646 | **<0.001** |
| 4 | 17 | 10.174 | **<0.001** | 9.927 | **0.001** |
| 5 | 15 | 13.066 | **<0.001** | 13.365 | **<0.001** |

[a]Dependent variable: IgA.
[b]$R = 0.354$, $R$ square $= 0.125$, adjusted $R$ square $= 0.109$, $F = 7.912$, $P < 0.001$.
[c]Represents partial regression coefficient.
[d]Refers to the reference group of a variable in the model. Bold text indicates that the differences observed are statistically significant.

**Multivariate regression analysis to explore the factors affecting IgG/IgA antibody levels.** We first explored the factors affecting IgG/IgA antibody levels in Omicron breakthrough cases using univariate and multivariate regression analyses. The results are shown in Table 4 and Table S2. The results of the multivariate regression showed that IgG/IgA antibody levels in Omicron breakthrough cases were mainly affected by the weeks of symptom onset; in particular, IgA antibody levels showed a linear increase as the weeks after symptom onset increased ($P < 0.001$). In addition, Table S3 shows that when the whole population was used as the study population, the <18 age group, the "booster dose" group, and the "acute/convalescent patient" group effectively increased the level of IgG antibody ($P < 0.05$). To further determine the role of IgG/IgA antibody levels in identifying breakthrough cases and postimmune populations, the receiver operating characteristic (ROC) curve analysis was performed. It was found that when the cutoff value of IgA was 1 S/CO, the sensitivity and specificity were 55.48% and 93.36%, respectively, with an area under the curve (AUC) of 0.744; when the cutoff value of IgG was 15 S/CO, the sensitivity and specificity were 67.98% and 93.22%, respectively, with an AUC of 0.806, further confirming the diagnostic value of both indicators. The detailed results are shown in Table S4.

## DISCUSSION

Immunity against SARS-CoV-2 infection is central to long-term control of the disease during the current pandemic (18). However, recent reports have shown that the number of documented cases of Omicron breakthroughs is increasing. Therefore, one of the key issues is to understand the magnitude and kinetics of humoral immunity to SARS-CoV-2 following an Omicron breakthrough infection.

Our intensive longitudinal sampling during Omicron BA.2 breakthrough infection revealed the early dynamics of specific antibody levels (IgG/IgM/IgA) against the ancestral Hu-1 SARS-CoV-2 S protein. In this study, a total of 456 acute and recovery serum samples from Omicron breakthrough cases were collected and compared with 693 serum samples

from fully vaccinated COVID-19-naive participants. The groups differed in basic demographic characteristics such as age and sex, but we corrected for these effects through multivariate regression analysis. The results showed that IgG/IgM/IgA positivity rates and antibody levels were highest at the recovery phase of the breakthrough cases, followed by the acute phase of breakthrough cases, and were lowest in the postimmune population. The three indicators showed a rapid increase in antibody levels in the first 2 weeks PSO in breakthrough cases, followed by a more stable range. In the postimmune population, there was a rapid decrease in antibody levels in the first 2 months following the last dose of vaccination, finally reaching a plateau. Furthermore, we calculated that the proportion of the ratio (IgG-recovery/IgG-acute) greater than or equal to four times was 50% and that the factor affecting IgG/IgA antibody levels in breakthrough cases was mainly the number of weeks since symptom onset.

Antibodies that bind to viral proteins contribute to the immune control of infection through increased clearance of free viruses or by targeting infected cells for immune clearance (19, 20). Some studies have shown that spike-specific IgG antibodies have an estimated half-life of 100 to 230 days (4, 21). In contrast, the IgM and IgA1 spike-binding antibodies have shorter half-lives of 55 and 42 days, respectively. Another study estimated an average half-life of 210 days for IgA over the first 8 months after SARS-CoV-2 infection (21). Therefore, residual SARS-CoV-2-specific nonneutralizing antibodies may confer protective benefits during reinfection, even when serum neutralizing activity declines below the threshold of protection (22). Several studies of COVID-19 patients have found that the level of specific IgM antibodies rises to near peak state in the second week after onset (23−26), with IgG antibodies appearing slightly later than IgM antibodies, and the positivity rate gradually rises to 43.5% to 76.0% in week 2 (27–29). IgA antibodies are distributed in the blood and mucous membranes and play an important role in local mucosal immunity (e.g., the respiratory epithelium) (30). The IgA positivity rate can exceed 50% in the first week after disease onset (25, 31). These findings differ slightly from those of the Omicron breakthrough cases in this study. What is consistent is that the positive rate for the three indicators was close to the peak in the second week after the onset of symptoms, and what is inconsistent is that the positive rate for IgG in breakthrough cases reached over 95% from the second week, which was higher than that in the infected group; however, the positive rate for IgA in breakthrough cases was only 34.10% in the first week, which was lower than that in the infected group. The results of Almendro-Vázquez et al. regarding specific humoral immunity against SARS-CoV-2 after natural infection also confirmed that IgG detection started on day 13 PSO and increased during convalescence (32). This conclusion was consistent with our findings. The comparison of antibody levels and positivity rates in the acute phase between the naturally infected population and immune breakthrough cases provides a more visual representation of the differences between the two, highlighting the role of vaccination, as well as filling a gap in the study of early antibody kinetics in Omicron breakthrough cases.

Encouragingly, we obtained the early kinetics (within 5 weeks of symptom onset) of specific antibodies (IgG/IgM/IgA) in Omicron breakthrough cases. Antibody levels of people who received inactivated vaccines were compared with antibody levels in the postimmune population who received two doses of inactivated vaccine, revealing that breakthrough cases produce a higher immune response than COVID-19-naive individuals. The reason for this phenomenon is that breakthrough infections are equivalent to secondary immunizations, activating the immune recall induced by previous vaccinations. Recalled immune memory can rapidly increase the serological titers of IgG, but not IgM, after antigen reexposure in vaccinated individuals. IgG is produced directly, without a class switch. Based on the results of the current analysis, infection can lead to further increases in IgG levels in the immunized population. These data lend further support for tracing or identifying previously infected individuals. When the real-time PCR results were negative, the difference in IgG and IgA levels was used to distinguish breakthrough cases from the immunized population. The results of the ROC curve analysis confirmed the reliability of this hypothesis. When the threshold values

for IgA and IgG were set to 1 and 15 S/CO, the AUCs to distinguish the difference between breakthrough cases and postimmunization populations were 0.744 and 0.806, respectively. In particular, the specificities were 93.36% and 93.22%, which means that it was very accurate to exclude a person as uninfected when the IgA and IgG were less than 1 and 15 S/Co, respectively. However, the sensitivity was relatively low to confirm a person as infected when the IgA and IgG were more than 1 and 15 S/CO, respectively. These immunological tests were mainly used in the epidemiological investigation. Fig. 2A shows that the IgG and IgA levels had an upward trend in the fourth week PSO in those breakthrough cases. Fig. 2B shows that the overall trend of IgG levels decreased, and IgA antibody levels fluctuated at low levels throughout the study period in the postimmune population. Therefore, our results were relatively stable when extrapolating infection history from a month ago, and this time is valuable and important for tracing the source in the epidemiological investigation.

When the population had no history of SARS-Cov-2 immunization, a 4-fold or higher increase in IgG antibody titers in the recovery phase compared to the acute phase was a criterion based on the Diagnostic and Treatment Guidelines for Novel Coronavirus Infection (trial tenth edition). It is worth investigating whether this general rule holds true after a history of inactivated vaccines. Therefore, this study validated the criterion by collecting serum samples from 46 individuals who had samples available from both the acute and recovery phases and found that 50% of the subjects met the guideline's criterion. This means that for individuals with a history of vaccination, if their IgG increased by four times in 2 to 4 weeks, it can be assumed that they were exposed to or infected with SARS-CoV-2. Of course, it would make more sense to work out a more accurate fold measure as a criterion for the inactivated vaccine population. This will require more samples and more in-depth research.

Finally, a highlight of this study was the multivariate regression analysis that included multiple independent variables to explore whether the key factors influencing antibody levels in these cases were the number of weeks since symptom onset and, to a lesser extent, the availability of booster immunizations. If the IgG level is around 36.3 S/CO (IgA, 3.1 S/CO), then the individual may be in the acute phase of the infection, and if the IgG level reaches 88.5 S/CO (IgA, 10.1 S/CO), then he or she may be in the convalescent phase of the infection, or 3 to 4 weeks postinfection. Since IgG/IgA levels rise to a plateau of higher values 2 weeks after infection, it is difficult to accurately determine the number of days of infection based on the specific value, but only a relative time range can be given. These data were analyzed from the perspective of population, and caution should be exercised when applying data to specific individuals due to individual differences.

Some important caveats of our study include the significant demographic differences between Omicron breakthrough cases and the postimmune population. In addition, we collected blood samples only from the early stages of Omicron breakthrough cases. These cases will need to be followed over a long period of time to understand the long-term kinetics of the specific antibodies. Furthermore, the antigen encapsulated in this assay was the wild-type strain of SARS-CoV-2, which showed some differences in amino acids compared with the S protein of the Omicron variant; however, the test results of antibody titers can be approximately substituted.

Overall, infection can lead to a further increase in IgG levels in the immunized population, and the ratio (IgG recovery/IgG acute) increases by a factor of 4. The titers of serological antibody still can play role in the auxiliary diagnosis of COVID-19 infection. When throat swab nucleic acid real-time PCR was negative, we used the kinetics of IgG and IgA levels to distinguish the breakthrough cases from the immunized population.

## MATERIALS AND METHODS

**Definition of study participants and data collection.** The study participants were categorized into two groups, i.e., individuals who were Omicron breakthrough cases post-inactivated vaccination and controls who were COVID-19-naive individuals with two doses of inactivated vaccination. A breakthrough infection was defined as the detection of SARS-CoV-2 on an reverse transcriptase PCR (RT-PCR) assay performed 14 or more days after vaccination of a second dose of CoronaVac/BBIBP. Omicron breakthrough cases were selected from individuals infected in Jiangsu Province, China, during the first half of 2022. Whole-genome sequencing (WGS) analysis determined all the samples of breakthrough cases to be subvariants of the Omicron variant BA.2. The control group ($n = 693$) was selected from voluntarily given blood samples collected in Changzhou, Jiangsu

Province, between 5 August and 8 August 2021. In total, 456 blood samples were collected to monitor the early kinetics of antibodies in breakthrough cases, with 308 blood samples collected during the acute phase (1 to 2 weeks post-symptom onset [PSO]) and 148 blood samples collected during the recovery phase (3 to 5 weeks PSO). Written informed consent was obtained from all participants or their legally authorized representatives. Clinical data included immunization history with the inactivated vaccine, demographics, and acute phase disease severity classification.

**Chemiluminescence immunoassay (CLIA)-based detection of specific antibodies against SARS-CoV-2.** In this study, sera were used to test the levels of the following antibodies: specific IgG, IgM, and IgA. These specific antibodies were tested using the following commercial kit: novel coronavirus (2019-nCoV) IgG/IgM/IgA antibody diagnostic kit (plate CLIA) supplied by Bioscience Co. (China National Medical Products Administration, approval numbers 20203400183 [IgG] and 20203400182 [IgM]) on an automated magnetic chemiluminescence analyzer (Axceed 260; Bioscience). The recombinant antigens contain the nucleoprotein and a peptide from the spike protein of the SARS-CoV-2 wild strain. Detailed information on the principle of the test, procedure, sensitivity, and specificity of this kit can be found in previous studies conducted by our team (15, 33). The chemiluminescent signal was measured as relative light units (RLU) using the analyzer's optical system. Antibody levels were presented as the measured chemiluminescence values divided by the cutoff (S/CO). The RLU of the samples was positively correlated with the concentration of IgM, IgG, and IgA antibodies (S/CO). The cutoff value of this test was defined by receiver operating characteristic curves (23). If the S/CO value is ≥1.0, the test result is positive, while if the S/CO value is <1.0, the result is considered negative. The detection performance of the magnetic particle chemiluminescent immunoassay (MCLIA) kits for specific antibodies was reported by the manufacturer, and the coincidence rate was 85 to 90%.

**Statistical analyses.** Categorical variables were expressed as percentages, and statistical significance was calculated using the chi-square or Fisher's exact test. Continuous variables were presented as the mean ± 95%CI or median ± IQR as appropriate, and significance was calculated either by applying a two-tailed unpaired $t$ test, one-way analysis of variance (ANOVA), the Wilcoxon rank sum (Mann-Whitney) test, or the Kruskal-Wallis test, as appropriate. Quantitative comparisons of IgG, IgM, and IgA levels between the two groups are presented as bar charts. The trend in the positivity rates or antibody levels over time for the three indicators is shown as a line plot. Univariate or multivariate regression analyses were used to explore the factors affecting antibody levels. Receiver operating characteristic (ROC) curve analysis was used to evaluate the ability of IgG and IgA to identify breakthrough cases in vaccinated populations. Stata (version 15.0) and GraphPad Prism (version 9.4) software were used for the statistical analysis. In all statistical analyses, a $P$ value of 0.05 was considered statistically significant.

## SUPPLEMENTAL MATERIAL

Supplemental material is available online only.

**SUPPLEMENTAL FILE 1**, DOCX file, 0.03 MB.

## ACKNOWLEDGMENTS

We are grateful for all patients who participated in this study. We also thank all the staff members who supported this study.

We declare that the research was conducted in the absence of any commercial or financial relationships that could be construed as a potential conflict of interest.

This work was supported by the Social Development Foundation of Jiangsu Province under grant no. BE2021739, the Postgraduate Research and Practice Innovation Program of Jiangsu Province under grant no. KYCX20_0153, the National Natural Science Foundation of China under grant no. 82041026, the Nanjing Important Science & Technology Specific Projects 2021-11005, the Wuxi Project of Science and Technology (no. Y20212042), the Top Talent Support Program for young and middle-aged people of Wuxi Health Committee (no. BJ2020100), and the Scientific Research Project of Jiangsu Health Commission (no. DX202301).

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
