## [Reviewer comments · Microbiology Spectrum]

Microbiology Spectrum

Unveiling a New Perspective on Distinguishing Omicron Breakthrough Cases and Post-immune COVID-19 Naïve Individuals: Insights from Antibody Profiles

Shihan Zhang, Chen Dong, Qian Zhen, Chao Shi, Hua Tian, Chuchu Li, Xiaoxiao Kong, Qi-Gang Dai, Haodi Huang, Aidibai Simayi, Fengcai Zhu, Yawen Xu, Jianli Hu, Ke Xu, Li-Ling Chen, Changjun Bao, Hui Jin, and Liguozhu

Corresponding Author(s): Hui Jin, Southeast University

Review Timeline:

Submission Date:	May 3, 2023
Editorial Decision:	May 21, 2023
Revision Received:	June 13, 2023
Accepted:	June 24, 2023

Editor: Rui Huang

Reviewer(s): Disclosure of reviewer identity is with reference to reviewer comments included in decision letter(s). The following individuals involved in review of your submission have agreed to reveal their identity: Naiying Mao (Reviewer #2)

Transaction Report:

DOI: <https://doi.org/10.1128/spectrum.01808-23>

May 21, 2023

Dr. Hui Jin
Southeast University
School of Public Health
Nanjing
China

Re: Spectrum01808-23 (Novel Role of Serum SARS-CoV-2 Specific IgG and IgA in Inferring the History and Duration of Omicron BA.2 Infection)

Dear Dr. Hui Jin:

Thank you for submitting your manuscript to Microbiology Spectrum. Your manuscript has been examined by two reviewers and the editors. We regret that it has not been accepted for publication in its current format. Although the reviews were encouraging, a number of concerns were raised that have prompted us to ask for some revisions. When submitting the revised version of your paper, please provide (1) point-by-point responses to the issues raised by the reviewers as file type "Response to Reviewers," not in your cover letter, and (2) a PDF file that indicates the changes from the original submission (by highlighting or underlining the changes) as file type "Marked Up Manuscript - For Review Only". Please use this link to submit your revised manuscript - we strongly recommend that you submit your paper within the next 60 days or reach out to me. Detailed instructions on submitting your revised paper are below.

Link Not Available

Sincerely,

Rui Huang

Journals Department
Reviewer comments:

Reviewer #1

Identifying breakthrough infection in inactivated virus vaccinees is very challenging when the nucleic acid-based diagnosis was found to be negative. The present study is a novel study since it adds more evidence to the role of the level and kinetics of binding antibodies in identifying breakthrough infection in individuals vaccinated with inactivated virus vaccine in whom. This study also provides the role of the ratio of the level of Recovery IgG titer to the level of Acute IgG titer not only identification of breakthrough infection but also in identifying acute infection. The major strength of this study is the inclusion of large cohorts (acute and recovery breakthrough cases and SARS-CoV-2 infection native vaccinees). However, the manuscript still needs

improvement. The claimed diagnostic role is difficult to implement when two serum/plasma samples collected at different time points from the same individual are unavailable.

Reviewer #2 (Comments for the Author):

This article by Shihan Zhang et al. analyzed the kinetic changes of specific IgG, IgM, and IgA antibodies to SARS-CoV-2 in breakthrough cases after inactivated vaccination and in COVID-19 naive individuals who were vaccinated with 2 doses of inactivated vaccine over time. The results of the study showed that Omicron breakthrough cases had significantly higher levels of IgG and IgA antibodies than the immunized population. It gives us a glimpse of the current status of SARS-CoV-2 reinfection after mass population vaccination, especially after a pandemic of Omicron variant strains. However, the findings of this report require some modifications before publication.

Major comments:

1. The name of article does not match the content enough, it is better to be changed.
2. Please clarify the case definition of the breakthrough cases in this study.
3. It is interesting to find in the article that the threshold values for IgA and IgG can be set to 1 and 15 S/CO for trying to distinguish the different between breakthrough cases and post-immunization populations, but the authors do not do a good job of explaining and analyzing this phenomenon in the discussion and whether it can be used for clinical diagnosis in the future. In particular, the duration of use of this threshold considers that antibody titers will decline over time.
4. Author calculated the proportion of IgG-recovery/IgG-acute ratios of four or more in the breakthrough cases with two blood collections in the acute and recovery phases and found that the proportion of ratios greater than four decreased as the level of IgG in the acute phase increased. However, considering that IgG antibodies in breakthrough cases rise much faster than in first-time infected patients, using 4-fold as a criterion may not be appropriate.
5. In the article the authors tried to use the level of IgG or IgA antibody values to infer the different infection stages infected cases were in, but this inference could lead to errors due to the big individual differences.

Minor comments:

1. In line 52, please explain whether the antibody level is the arithmetic mean or median?
2. In line 174, how many post-immune individuals received at least one booster immunization and what is the impact on the result.

Staff Comments:

Preparing Revision Guidelines

Please return the manuscript within 60 days; if you cannot complete the modification within this time period, please contact me. If you do not wish to modify the manuscript and prefer to submit it to another journal, please notify me of your decision immediately so that the manuscript may be formally withdrawn from consideration by Microbiology Spectrum.

Reviewer #1:

Identifying breakthrough infection in inactivated virus vaccinees is very challenging when the nucleic acid-based diagnosis was found to be negative. The present study is a novel study since it adds more evidence to the role of the level and kinetics of binding antibodies in identifying breakthrough infection in individuals vaccinated with inactivated virus vaccine in whom. This study also provides the role of the ratio of the level of Recovery IgG titer to the level of Acute IgG titer not only identification of breakthrough infection but also in identifying acute infection. The major strength of this study is the inclusion of large cohorts (acute and recovery breakthrough cases and SARS-CoV-2 infection native vaccinees). However, the manuscript still needs improvement. The claimed diagnostic role is difficult to implement when two serum/plasma samples collected at different time points from the same individual are unavailable.

Response: We appreciate you very much for your positive and constructive comments and suggestions on our manuscript. In response to your mentioned implementability issue, in the prevention and control of the pandemic, we generally collect samples twice from suspected subjects, with a general interval of 1~2 weeks between samples. When the first sampling is finished, timely testing is done to apply quantitative values of IgG and IgA to roughly determine whether the host has a history of infection. When the antibody results of the second sampling are available, further verification will be done.

Reviewer #2:

This article by Shihan Zhang et al. analyzed the kinetic changes of specific IgG, IgM, and IgA antibodies to SARS-CoV-2 in breakthrough cases after inactivated vaccination and in COVID-19 naive individuals who were vaccinated with 2 doses of inactivated vaccine over time. The results of the study showed that Omicron breakthrough cases had significantly higher levels of IgG and IgA antibodies than the immunized population. It gives us a glimpse of the current status of SARS-CoV-2 reinfection after mass population vaccination, especially after a pandemic of Omicron variant strains. However, the findings of this report require some modifications before publication.

Major comments:

1. The name of article does not match the content enough, it is better to be changed.

Response: Thank you for your suggestion. We have revised the title to: "**Unveiling a New Perspective on Distinguishing Omicron Breakthrough Cases and Post-immune COVID-19 Naïve Individuals: Insights from Antibody Profiles**".

2. Please clarify the case definition of the breakthrough cases in this study.

Response: Thank you for your suggestions, and we sincerely apologize for our oversight during the writing process. We have now added a precise definition of breakthrough cases as the detection of SARS-CoV-2 on RT-PCR assay performed 14 or more days after vaccination of a second dose of CoronaVac/BBIBP in lines 117-119 of the manuscript.

3. It is interesting to find in the article that the threshold values for IgA and IgG can be set to 1 and 15 S/CO for trying to distinguish the different between breakthrough cases and post-immunization populations, but the authors do not do a good job of explaining and analyzing this phenomenon in the discussion and whether it can be used for clinical diagnosis in the future. In particular, the duration of use of this threshold considers that antibody titers will decline over time.

Response: Thank you for your careful review. When the threshold values for IgA and IgG were set to 1 and 15 S/CO, the AUC to distinguish the different between breakthrough cases and post-immunization populations were 0.744 and 0.806, respectively. Especially, the specificity were 93.36% and 93.22%, which means that it was of high accuracy to exclude one as an uninfected person when the IgA and IgG was less than 1 and 15 S/CO. However, the sensitivity were relatively low to confirm one as an infected person when the IgA and IgG was more than 1 and 15 S/CO. These immunological tests were mainly used in the course of epidemiological investigation and were not necessarily suitable for clinical diagnosis.

Figure 2A showed that the IgG and IgA level had an upward trend in the fourth week of PSO in those breakthrough cases. Figure 2B showed that the overall trend of IgG levels decreased, and IgA antibody levels fluctuated at low levels throughout the study period in the post-immune population. Therefore, our results were relatively stable when extrapolating infection history from a month ago, and this time is valuable and important for tracing the source in the epidemiological investigation.

We have now modified it in lines 319-331 of the manuscript.

4. Author calculated the proportion of IgG-recovery/IgG-acute ratios of four or more in the breakthrough cases with two blood collections in the acute and recovery phases and found that the proportion of ratios greater than four decreased as the level of IgG in the acute phase increased. However, considering that IgG antibodies in breakthrough cases rise much faster than in first-time infected patients, using 4-fold as a criterion may not be appropriate.

Response: Thank you for the suggestion. When the population were with no history of SARS-CoV-2 immunization, a 4-fold or higher increase in IgG antibody titers in the recovery phase compared to the acute phase was a criterion based on the Diagnostic and Treatment Guidelines for Novel Coronavirus Infection (Trial Tenth Edition). It is worth investigating whether this general rule holds true after a history of

inactivated vaccines. Therefore, this study validated the criterion by collecting serum samples from 46 individuals who had samples from both the acute and recovery phases, and found that 50% of the subjects met the guideline's criterion. Of course, it would make more sense to work out a more accurate fold as a criterion for the inactivated vaccine population. This will require more samples and more in-depth research.

We have now modified it in lines 332-344 of the manuscript.

5. In the article the authors tried to use the level of IgG or IgA antibody values to infer the different infection stages infected cases were in, but this inference could lead to errors due to the big individual differences.

Response: Thank you for your question. In our study, by univariate and multivariate regression analysis, based on factors such as age, sex and vaccination history, we found that IgG and IgA antibody levels in Omicron breakthrough cases were mainly affected by the weeks of symptom onset. These were analyzed from the perspective of population, and caution should be exercised when applied to specific individuals due to the individual differences. We have now modified it in lines 353-355 of the manuscript.

Minor comments:

1. In line 52, please explain whether the antibody level is the arithmetic mean or median?

Response: After verification, the mentioned antibody levels in the manuscript refer to arithmetic means.

2. In line 174, how many post-immune individuals received at least one booster immunization and what is the impact on the result.

Response: As shown in Table 1, all post-immune naïve individuals received two doses of inactivated vaccine, and no booster immunizations were administered.

June 24, 2023

Dr. Hui Jin
Southeast University
School of Public Health
Nanjing
China

Re: Spectrum01808-23R1 (Unveiling a New Perspective on Distinguishing Omicron Breakthrough Cases and Post-immune COVID-19 Naïve Individuals: Insights from Antibody Profiles)

Dear Dr. Hui Jin:

Your manuscript has been accepted, and I am forwarding it to the ASM Journals Department for publication. You will be notified when your proofs are ready to be viewed.

Sincerely,

Rui Huang
Editor, Microbiology Spectrum
